# Exposure to Secondhand Smoke Extract Increases Cisplatin Resistance in Head and Neck Cancer Cells

**DOI:** 10.3390/ijms25021032

**Published:** 2024-01-14

**Authors:** Balaji Sadhasivam, Jimmy Manyanga, Vengatesh Ganapathy, Pawan Acharya, Célia Bouharati, Mayilvanan Chinnaiyan, Toral Mehta, Basil Mathews, Samuel Castles, David A. Rubenstein, Alayna P. Tackett, Yan D. Zhao, Ilangovan Ramachandran, Lurdes Queimado

**Affiliations:** 1Department of Otolaryngology Head and Neck Surgery, The University of Oklahoma Health Sciences Center, Oklahoma City, OK 73104, USA; balaji-sadhasivam@ouhsc.edu (B.S.); jmanyanga@cytovance.com (J.M.); vengatesh-ganapathy@ouhsc.edu (V.G.); mayilvanan-chinnaiyan@ouhsc.edu (M.C.); toral.mehta@osumc.edu (T.M.); bmathews@bva20-20.com (B.M.); scastles@southcentralfoundation.com (S.C.); 2Department of Occupational and Environmental Health, The University of Oklahoma Health Sciences Center, Oklahoma City, OK 73104, USA; 3Department of Cell Biology, The University of Oklahoma Health Sciences Center, Oklahoma City, OK 73104, USA; 4Department of Biostatistics, The University of Oklahoma Health Sciences Center, Oklahoma City, OK 73104, USA; pawan-acharya@ouhsc.edu (P.A.); daniel-zhao@ouhsc.edu (Y.D.Z.); 5Department of Biomedical Engineering, Stony Brook University, New York City, NY 11794, USA; david.rubenstein@stonybrook.edu; 6TSET Health Promotion Research Center, Stephenson Cancer Center, The University of Oklahoma Health Sciences Center, Oklahoma City, OK 73104, USA; alayna.tackett@osumc.edu; 7Division of Medical Oncology, The Ohio State University, Columbus, OH 43210, USA; 8Department of Endocrinology, Dr. ALM Post Graduate Institute of Basic Medical Sciences, University of Madras, Taramani Campus, Chennai, TN 600113, India; ilangovan@unom.ac.in

**Keywords:** head and neck squamous cell carcinoma (HNSCC), oral cancer, response to therapy, secondhand smoke, IC_50_, multidrug resistance, cisplatin, cell death

## Abstract

Chemotherapy and radiotherapy resistance are major obstacles in the long-term efficacy of head and neck squamous cell carcinoma (HNSCC) treatment. Secondhand smoke (SHS) exposure is common and has been proposed as an independent predictor of HNSCC recurrence and disease-free survival. However, the underlying mechanisms responsible for these negative patient outcomes are unknown. To assess the effects of SHS exposure on cisplatin efficacy in cancer cells, three distinct HNSCC cell lines were exposed to sidestream (SS) smoke, the main component of SHS, at concentrations mimicking the nicotine level seen in passive smokers’ saliva and treated with cisplatin (0.01–100 µM) for 48 h. Compared to cisplatin treatment alone, cancer cells exposed to both cisplatin and SS smoke extract showed significantly lower cisplatin-induced cell death and higher cell viability, IC_50_, and indefinite survival capacity. However, SS smoke extract exposure alone did not change cancer cell viability, cell death, or cell proliferation compared to unexposed control cancer cells. Mechanistically, exposure to SS smoke extract significantly reduced the expression of cisplatin influx transporter CTR1, and increased the expression of multidrug-resistant proteins ABCG2 and ATP7A. Our study is the first to document that exposure to SHS can increase cisplatin resistance by altering the expression of several proteins involved in multidrug resistance, thus increasing the cells’ capability to evade cisplatin-induced cell death. These findings emphasize the urgent need for clinicians to consider the potential role of SHS on treatment outcomes and to advise cancer patients and caregivers on the potential benefits of avoiding SHS exposure.

## 1. Introduction

Tobacco smoking is the leading cause of preventable disease and death, causing over 7 million deaths worldwide annually [1]. Tobacco smoke contains more than 7000 chemicals, 70 of which are known carcinogens [2]. Tobacco smoke can be divided into mainstream smoke, the main smoke inhaled by active smokers, and sidestream (SS) smoke, the smoke released from the smoldering cigarette tip. SS smoke accounts for about 85% of secondhand smoke (SHS) [3]. Mainstream and SS smoke have the same chemicals [4]. However, compared to mainstream smoke, SS contains higher levels of nicotine and carcinogens, such as polycyclic aromatic hydrocarbons (PAHs), nitrosamines, and aromatic amines [4,5,6]. Over 20% of nonsmoking U.S. adults are exposed to SHS. The prevalence of SHS exposure is highest (~40%) among non-Hispanic blacks, persons living in poverty, and children aged 3–11 years [7,8]. SHS exposure causes more than 41,000 deaths among nonsmoking adults and 900 deaths in infants each year in the U.S. [9]. Meta-analysis studies support a causal association between SHS exposure and diverse cancers in never-smokers [10], including lung adenocarcinoma [11] and oral cancer [12].

Head and neck squamous cell carcinoma (HNSCC) is the sixth most prevalent cancer worldwide, with 890,000 new cases and 450,000 deaths in 2018 [13]. Tobacco smoking is one of the most important risk factors for head and neck cancer [14]. The risk for head and neck cancer is about 10-fold higher in smokers than in never-smokers [15]. Moreover, being a current smoker at the time of HNSCC diagnosis is well established as a negative prognostic factor [16]. Over 70% of newly diagnosed HNSCC patients present with advanced disease require multimodality chemotherapy and radiation with or without surgery [17]. These intensive treatments have significant debilitating and disfiguring consequences. Patients frequently experience a loss of aesthetics and difficulties in chewing, swallowing, speaking, and breathing [15]. The paramount importance of the head and neck structures and the dramatic physical and social consequences of HNSCC multimodality therapy make these tumors particularly difficult to treat. Notably, late diagnosis and therapy resistance are the major factors contributing to the poor prognosis of HNSCC. 

Cisplatin, an intercalating agent, and its derivative compounds are essential tools to treat a wide range of cancers. Cisplatin is the preferred drug for HNSCC primary systemic therapy with concurrent radiotherapy and for primary chemotherapy with postoperative chemoradiation [18]. Despite a very high rate of initial responses, the development of chemoresistance is frequent and leads to therapeutic failure. Continued tobacco smoking is common in over 50% of HNSCC patients, is associated with increased cisplatin resistance [19,20], and is an independent prognostic factor for HNSCC overall survival [21,22,23]. To the best of our knowledge there are no reports on the effects of SHS exposure on cancer therapy efficacy. Nonetheless, a single prospective cohort study has shown that SHS exposure is an independent predictor of HNSCC recurrence and disease-free survival [24]. Both recurrence and disease-free survival reflect treatment efficacy. To better understand the impact of SHS exposure on HNSCC chemotherapy response and recurrence, we investigated the effects of SHS exposure on HNSCC cisplatin treatment efficacy and clonogenic survival. We have also dissected the potential mechanisms by which SHS exposure contributes to HNSCC cisplatin resistance.

## 2. Results

### 2.1. SS Smoke Extract Increases the Cell Viability of Cisplatin-Treated HNSCC Cells 

To mimic SHS exposure, HNSCC cells were exposed to SS extracts delivering 48 ng/mL nicotine. This value was chosen to match the reported nicotine average in the saliva of non-smokers exposed to secondhand smoke [25]. Cells were treated with cisplatin for 48 h, in the presence or absence of SS smoke extract. In the toxicology field, a 48 h exposure is standard for measuring the impact of tobacco smoke in many cellular processes, including cisplatin resistance [20]. We observed that all three HNSCC cell lines exposed to SS smoke extract for 48 h followed by treatment with cisplatin for 48 h in the presence of SS smoke extract showed a significant increase in cell viability when compared to cells not exposed to SS smoke extract (Figure 1): UM-SCC-1 (*p* = 0.007), WSU-HN6 (*p* = 0.002), and WSU-HN30 (*p* = 0.0001). Linear regression analysis adjusting for cell lines, further strengthened our observation that SS smoke exposure increased the cell viability of cisplatin-treated cancer cells across all the cell lines studied (*p* < 0.0001). 

### 2.2. SS Smoke Extract Alone Does Not Alter HNSCC Cell Viability, Cell Death, or Cell Proliferation

These observations could reflect an increase in cell viability induced by SS smoke alone or a reduction in cisplatin-induced cell death in the presence of SS smoke. Thus, first we investigated the effect of SS smoke exposure on HNSCC cell’s viability using MTT assay. Upon exposure of HNSCC cells to SS smoke extract alone for up to 96 h, we observed no significant change in cell viability at various time points (0, 24, 48, 72, and 96 h) compared to unexposed cancer cells (control group) in each of the three cell lines studied (Figure 2A–C). Consistently, we observed no significant change in cell death, as assessed by Incucyte cytotox red dye, at various time points (0, 24, 48, 72, and 96 h) upon exposure to SS smoke extract alone (Figure 2D–F). 

Moreover, the number of proliferating cells, as assessed by Ki67 staining, was similar between SS smoke extract exposed and unexposed control cells in all three cell lines (Figure 3A–C). 

Altogether, these data show that exposure to SS only does not change cell viability, cell death, or cell proliferation. Thus, our data suggest that the observed increase in cell viability (depicted in Figure 1) reflects an interference of SS smoke on cisplatin action.

### 2.3. SS Smoke Extract Reduces Cisplatin-Induced HNSCC Cell Death and Apoptosis 

Having documented that the observed increase in cell viability was not due to the presence of SS smoke alone, we used live cell imaging (Incucyte^®^ Live-Cell analysis) and continuous documentation to measure the cisplatin-induced cell death in the presence or absence of SS smoke extract. We observed that, when compared to cisplatin-alone-treated cells, the presence of SS smoke extract led to a significant decrease in cisplatin-induced cell death (*p* < 0.0001) in all three cell lines (Figure 4A–C). Notably, this effect was significant at every time point tested between 27 and 45 h of cisplatin treatment for all three cell lines (Figure 4A–C). Thus, we investigated whether the presence of SS smoke also reduced cisplatin-induced apoptosis. The Incucyte^®^ Live-Cell analysis assay for apoptosis revealed that SS extract exposure significantly decreased cisplatin-induced apoptosis in UM-SCC-1 (*p* < 0.0001), WSU-HN6 (*p* < 0.05), and WSU-HN30 (*p* < 0.0001) cells (Figure 4D–F). These data suggest that SS smoke exposure blocks cisplatin-induced apoptosis leading to cisplatin resistance. Linear regression analysis comparing the difference between cisplatin alone and cisplatin in the presence of SS, after adjusting all three cell lines’ data points from Figure 4D–F as a sum, further strengthened our observation that SS smoke exposure decreased apoptosis across all the cell lines studied (*p* < 0.0001).

Altogether, these data show that during cisplatin treatment, exposure to SS smoke extract increases cell viability by reducing cisplatin-induced cancer cell death, which may lead to poor cisplatin treatment efficacy.

### 2.4. SS Smoke Extract Exposure Increases Cisplatin IC_50_ in HNSCC Cells 

To further measure the impact of SS smoke on cisplatin efficacy, we calculated the half-maximal inhibitory concentration (IC_50_) in the presence and absence of SS smoke extract. The dose-dependent symmetrical sigmoidal curves (Figure 5A) showed that the concentration of cisplatin required to induce a 50% reduction in cell viability was significantly higher in the presence of SS smoke than in cells not exposed to SS (treated with cisplatin alone) for all three cell lines tested: UM-SCC-1 (*p* < 0.0001), WSU-HN6 (*p* = 0.0035), and WSU-HN30 (*p* < 0.0001). Our data show that in presence of SS smoke extract, approximately twice the concentration of cisplatin is required to achieve the same treatment efficacy (Figure 5B). These data show that exposure to SS increases resistance to cisplatin treatment in vitro. Since cisplatin IC_50_ is an in vitro marker predictive of cisplatin response in cancer patients [26], these data suggest that exposure to SHS during cancer treatment might reduce cisplatin efficacy. 

### 2.5. SS Smoke Extract Exposure during Cisplatin Treatment Increases the Clonogenic Survival of Head and Neck Cancer Cells

The ability of cancer cells to continue to proliferate after chemotherapy leads to cancer recurrence, invasion, and ultimately metastasis. To determine how exposure to SS smoke impacts the long-term ability of tumor cells to grow after cisplatin treatment, we measured the clonogenic survival of head and neck cancer cells. Upon 48 h of cisplatin treatment, we observed a significant increase in colony formation in SS-smoke-exposed UM-SCC-1 (*p* = 0.026), WSU-HN6 (*p* < 0.001), and WSU-HN30 (*p* < 0.001) cells compared to unexposed control cells (Figure 6A,B). A significant increase in colony formation in the presence of SS smoke extract was also observed upon 24 h cisplatin treatment (*p* < 0.01, for all three HNSCC cell lines) (Appendix A), further suggesting that SS smoke exposure increases clonogenic survival in all the cell lines studied. 

These data show that exposure to SS smoke extract during cisplatin treatment significantly increases HNSCC clonogenic survival, a measure of indefinite reproductive ability [27].

### 2.6. SS Smoke Exposure Does Not Upregulate the Main DNA Repair Pathways Involved in Cisplatin Adduct Reduction

Cisplatin toxicity is determined by the formation of bulky intra- and inter-strand DNA crosslinks, which ultimately lead to cell death [28]. One of the main mechanisms by which cancer cells develop cisplatin resistance is the upregulation of the DNA repair pathways essential to remove these bulky lesions and to fix the double-strand breaks (DSBs) that are a byproduct of inter-strand crosslink repair. Of note, intra-strand cross links constitute less than 5% of all DNA lesions induced by cisplatin [29,30]. To evaluate whether an increase in repair was induced by the presence of SS smoke extract, we first assessed the effect of SS smoke extract on nucleotide excision repair, the main mechanism that detects and removes the cisplatin-induced DNA adducts [29,30,31]. Our analysis focused on *ERCC1*, *MMS19*, and *XPA*, the three main nucleotide excision repair genes that have been associated with response to cisplatin. Real-time RT-PCR data showed that, compared to unexposed control cells, *ERCC1* and *MMS19* mRNA expression were significantly decreased after SS smoke exposure in all three cell lines (Figure 7A,B), whereas *XPA* mRNA expression was decreased in UM-SCC-1 and WSU-HN30 cells, but increased in WSU-HN6 cells (Figure 7C). 

ERCC1 has been shown to participate in both nucleotide excision repair and homologous recombination. Thus, we next used γH2AX staining to investigate whether exposure to SS smoke extract for up to 96 h increases the formation of DSBs. γH2AX staining showed no significant increase in the number of cells with DSBs or the number of DSBs per cell between control and SS-smoke-extract-exposed HNSCC cells at 48 h (Figure 8A,B) or at 96 h (Figure 8C,D & Appendix A). 

Altogether, these data strongly suggest that, rather than an increase in DNA repair, an alternative mechanism is responsible for the observed increase in cisplatin resistance induced by exposure to SS smoke extract.

### 2.7. SS Smoke Extract Exposure Alters the Expression of Multidrug-Resistant Proteins 

Besides upregulation of DNA repair pathways, reduction in intracellular cisplatin availability is the major mechanism contributing to cisplatin resistance [28]. Thus, we assessed the effect of SS exposure on the expression of multidrug transporter proteins with a main role in cisplatin resistance. Compared to unexposed control cells, a significant increase in cisplatin efflux transporters ABCG2 (*p* ≤ 0.01) and ATP7A (*p* < 0.01) was observed in all the cell lines (Figure 9A,B). In contrast, a significant decrease in the cisplatin influx protein CTR1 was observed in all the cell lines tested (*p* < 0.01) (Figure 9A,B). These data suggest that SS smoke exposure increases cisplatin resistance by reducing cisplatin cellular availability, possibly through both reduced cisplatin uptake and increased cisplatin efflux.

To assess whether the observed alterations in protein expression reflected a pre- or post-translational effect of SS smoke exposure, we quantified the mRNA expression for the respective genes (Figure 9C). Consistent with the observed increase in protein expression, following SS smoke exposure, we observed a significant increase in *ABCG2* mRNA expression in all three cell lines, and in *ATP7A* mRNA expression in UM-SCC-1 and WSU-HN6. Despite a significant increase in ATP7A protein, *ATP7A* mRNA levels did not change in WSU-HN30 cells. There was no correlation between *CTR1* mRNA and protein expression. Our data suggest that SS smoke extract exerts both a transcriptional and post-transcriptional effect in diverse genes that contribute to cisplatin resistance.

## 3. Discussion 

Using well-established in vitro cancer cell line models to study the biological responses to therapy [32] and assays that have been proven to be predictive of cancer chemoresistance in patients [26], the present study shows for the first time that exposure to SS smoke reduces HNSCC cisplatin-induced cell death and increases cisplatin resistance by potentially decreasing the cisplatin cellular influx and increasing the cisplatin cellular efflux. In cancer patients, an increase in cancer resistance to chemotherapy or radiotherapy increases tumor recurrence and reduces disease-free survival [29]. Thus, our data provide unique and strong scientific support to the reported prospective study documenting that SHS exposure is an independent predictor of recurrence and disease-free survival after head and neck cancer treatment [24]. Both recurrence and disease-free survival reflect treatment efficacy. In the current study, we exposed HNSCC cells to SS smoke extract delivering 48 ng/mL nicotine, a concentration similar to that reported in the saliva of non-smokers 2 h after exposure to SHS [25] and within the observed range for adults and children exposed to SHS [33,34]. Despite smoke-free indoor air laws and reductions in combustible cigarette use [35], about 68.5 million adult Americans and 38% of children aged 3–11 years are exposed to SHS yearly [7,8]. Recent studies reported mean cotinine saliva levels of 18.3 ng/mL in passive adult smokers [33] and median cotinine saliva levels of 5.3 ng/mL (IQR = 2.3–9.1; max 28.8) in children with a potentially SHS-related illness [34]. Saliva nicotine levels are on average 10–20 times higher than in serum [25,36], and 5–10 times higher than cotinine [25,37]. Thus, our observations have significant implications for the treatment of both adults and children with cancer who are exposed to SHS. 

To the best of our knowledge, we are the first to document that exposure to SS smoke decreases HNSCC cisplatin-induced cell death, and increases HNSCC cell viability, cisplatin IC_50_ value, and clonogenic survival. The observed SS smoke-induced increase in cell viability in cisplatin-treated HNSCC cells was due to a reduction in cisplatin-induced cell death, rather than an increase in cell proliferation. These results are consistent with the previous report that SS smoke exposure did not increase proliferation, but increased cell survival [38]. In contrast, studies that used individual components of SS smoke, such as nicotine, benzo(a)pyrene, formaldehyde, or nicotine-derived nitrosamines observed an increase in cell proliferation, rather than a decrease in cell death [39,40]. Whether this difference is due to the use of pure chemicals by these last studies or the fact that Wong et al. [38] and the current study exposed cells to relatively low SS smoke levels is unknown. 

We showed that the presence of SS smoke extract causes an approximately two-fold increase in IC_50_ for each of the cell lines tested. IC_50_ values are the most commonly accepted way to assess drug efficacy in the lab [26,41], have been reported to be predictive of cancer chemoresistance in patients [26], and have been integrated into in vitro high-throughput drug screenings both to exclude and to identify additional therapies for high-risk cancer patients [41]. Thus, the observed increase in cisplatin IC_50_ in the presence of SS suggests that head and neck cancer patients exposed to SHS might require higher cisplatin doses than patients not exposed to SHS to attain a complete response to therapy. However, higher cisplatin doses might not be feasible due to organ toxicity. The presence of SS smoke during cisplatin treatment also led to a significant increase in clonogenic survival in all the cell lines and at all the time points tested (24 and 48 h of cisplatin treatment) which reflects unlimited proliferative capacity after cisplatin treatment. These observations are consistent with a recent and sole publication negatively linking SHS exposure to HNSCC prognosis [24]. Although the data on SHS impact on patient prognosis are scarce, active tobacco smoking at the time of HNSCC diagnosis has been consistently associated with lower rates of complete response to platinum-based induction chemotherapy and radiotherapy [23,42,43]. HNSCC patients who are active smokers during radiation therapy also have significantly lower locoregional control, disease-free survival, and five-year overall survival, than former smokers or never-smokers [43,44]. Given the similarity of the composition of mainstream and SS smoke, our data strongly suggest that like active smoking, exposure to SHS during HNSCC patient’s cisplatin treatment may reduce the therapeutic efficacy and lead to poor prognosis.

Over 900 proteins have been associated with platinum resistance [45]. Yet, there is consensus that the formation of DNA platinum adducts is the determinant step in cisplatin toxicity, leading to cell cycle arrest and/or apoptosis [28]. Thus, decreased intracellular cisplatin levels, increased repair of cisplatin-induced DNA lesions, and elevated apoptosis thresholds have been identified as common mechanisms contributing to cisplatin resistance [28,45,46]. In the present study, HNSCC cells exposed to SS smoke significantly reduced the expression levels of nucleotide excision repair genes. These data are not entirely surprising as SS exposure has been previously shown to reduce both nucleotide excision repair and base excision repair in lung tissue [47]. Similarly, we have previously reported that exposure to mainstream smoke also decreases the expression of ERCC1 protein [19]. At the physiologically relevant SS smoke extract dose used in the current study, we did not observe a change in the number of DSBs in cells exposed to SS smoke extract compared to unexposed cells. Although we have not measured overall DNA damage in this study, we and others have previously reported that exposure of human cells to SS smoke extract (at the levels present in the plasma of individuals exposed to SHS) led to a small but significant increase in overall DNA damage [48,49]. Whether this increase in DNA damage reflects mostly the presence of genotoxics in SS smoke or a decrease in DNA repair activity is beyond the scope of this study. Nonetheless, altogether, these data suggest that the observed increase in cisplatin resistance induced by SS smoke exposure is not due to an increase in DNA damage repair. 

One of the most important mechanisms of cisplatin resistance is a reduction in cisplatin cellular availability [29]. We observed that SS smoke exposure decreased CTR1 protein, a cisplatin influx transporter, and increased cisplatin efflux transporter ATP7A and multidrug-resistant ABCG2 proteins in all the cell lines tested. Both, low CTR1 and high ATP7A protein expression are associated with low intracellular cisplatin availability, and have been shown to be predictive of cisplatin resistance [28,50]. Both, CTR1 and ATP7A, also transport copper with high affinity. As expected, copper chelators have also been shown to enhance cisplatin efficacy [51]. ABCG2 extrudes diverse cellular compounds including cisplatin and tobacco carcinogens [52]. Cigarette smoke has been shown to increase ABCG2 expression in diverse cancer cells [53]. In head and neck cancer, ABCG2 protein expression correlates with pack-years of tobacco use and cisplatin resistance [20]. Altogether, our data suggest that exposure to SS smoke, the main component of SHS, reduces cisplatin-induced cell death and increases cisplatin resistance by reducing intracellular cisplatin availability, and therefore, might have significant influence on therapeutic outcomes. Although highly supported by prospectively collected data showing that SHS exposure is an independent predictor of HNSCC recurrence and disease-free survival [24], our study has the limitation of being an in vitro study. Studies measuring SHS exposure among cancer patients and their response to chemotherapy-based regimens are essential to measure the precise impact of SHS on cancer therapy outcomes. Additional mechanistic studies are also needed to fully understand the effects of SHS exposure on head and neck cancer resistance to therapy.

In conclusion, exposure to SS smoke, the main component of SHS, at levels similar to current human exposure, alters the expression of multidrug-resistant transporters and reduces cisplatin-induced cell death leading to an increase in cisplatin resistance in head and neck cancer cells. Furthermore, SS exposure during cisplatin treatment increases the clonogenic survival of cancer cells, which is essential for cancer recurrence, invasion, and ultimately, metastasis. These findings have critical clinical implications and provide a novel mechanism for the observed increase in HNSCC recurrence in patients exposed to SHS. To the best of our knowledge, this is the first study to report on the potential negative effects of SHS exposure during cancer chemotherapy. Such knowledge is crucial in advising cancer patients and caregivers in order to achieve the best chemotherapeutic outcomes.

## 4. Materials and Methods

### 4.1. Cell Culture 

Human HNSCC cell lines (UM-SCC-1, WSU-HN6, WSU-HN30) were cultured in Dulbecco’s modified Eagle’s Medium (DMEM) with 10% fetal bovine serum, under standard conditions [54]. The WSU-HN6 and WSU-HN30 cell lines were established at Wayne State University (primary source Dr. John Ensley) [55]. The UM-SCC-1 cell line was established at the University of Michigan (primary source Dr. Thomas E. Carey) [56]. All cell lines were verified and authenticated yearly by STR-based DNA profiling and multiplex PCR with CellCheck Cell Line Authentication (IDEXX BioResearch, Columbia, MO, USA).

### 4.2. Tobacco Smoke Extracts and Cisplatin Preparation 

Sidestream (SS) tobacco smoke extract was prepared from Marlboro 100s (16 mg of tar and 1.2 mg nicotine) cigarettes, following a two 50 mL puffs per minute topography regimen, and standard collection and storage protocols [49,57]. Nicotine concentration was determined using Gas Chromatography Mass Spectroscopy (GCMS) [58]. SS smoke extracts were added directly to the culture medium, to deliver 48 ng/mL nicotine. Sterile HEPES buffer saline was used as control. Cisplatin was reconstituted in 0.9% *w*/*v* sodium chloride to a stock concentration of 1 mg/mL and kept at −20 °C. The desired concentrations were obtained by serial dilution in culture media. 

### 4.3. Cell Viability and Cell Proliferation Analysis 

Cell viability was assessed using MTT assay as we previously described [19]. Briefly, cells were plated overnight in 96-well plates, and then exposed to SS smoke extract for 48 h followed by treatment with cisplatin (10 µM) for another 48 h, in the presence or absence of SS smoke extract. Cell viability was determined at 96 h. Experiments were repeated independently at least three times, each with 4 to 6 replicates. To assess cell proliferation, cells were plated overnight in 8-well chambered slides (Celltreat, Pepperell, MA, USA) and then cultured for an additional 96 h in the presence or absence of SS extract. Cell proliferation marker Ki67 was evaluated by immunofluorescence as per manufacturer’s protocol (Cell Signaling Technology, Danvers, MA, USA). ProLong™ Diamond Antifade Mountant with DAPI (Invitrogen, Waltham, MA, USA) was used to stain the nucleus. Experiments were repeated independently twice with duplicate wells. Ki67-positive cells were counted on at least 6 different fields per experiment, each with a minimum of 100 cells. Proliferation results are expressed as a percentage of the total number of cells. 

### 4.4. Cell Death and Apoptosis Analysis 

To measure the cell death and apoptosis, we used the Incucyte^®^ S3 Live-Cell analysis system. Briefly, for cell death quantification, cells were plated in 96-well plates, and cultured with Incucyte cytotox (red or green) dye (1:4000 dilution, Sartorius, Goettingen, Germany) in the presence or absence of SS extract for 48 h, followed by another 48 h in the presence of Incucyte cytotox (red or green) dye, with or without SS smoke extract, and with or without cisplatin treatment (10 µM). The total number of apoptotic cells was counted using Incucyte Caspase-3/7 (red or green) dye for apoptosis (1:500 dilution, Sartorius, Goettingen, Germany) as per manufacturer’s protocol. For all experiments, cells were placed in Incucyte^®^ S3 Live-Cell Analysis Instrument (Sartorius, Goettingen, Germany), which is inside the incubator (37 °C with 5% CO_2_) throughout the study. Images were taken every 3 h, and dead cells were counted using Incucyte 2020A software (version v2022B). Each well was imaged for a minimum of 4 different fields at each time point. Duplicate wells were used for each treatment group. Experiments were repeated independently twice.

### 4.5. Cisplatin Chemosensitivity Assay 

Cisplatin chemosensitivity was performed as we previously described [19]. Briefly, cells were seeded overnight, incubated for 48 h with or without SS smoke extract, followed by an additional 48 h of treatment with cisplatin (0, 0.1, 1, 2.5, 5, 10, 25, 50 and 100 µM) in the presence or absence of SS smoke extract. Cell viability was assessed using MTT assay. Data were analyzed using GraphPad Prism software (GraphPad, San Diego, CA, USA; version 9.0.0) to determine the half-maximal inhibitory concentration (IC_50_) values for each cell line. Experiments were repeated at least three times with minimum of 4 wells. 

### 4.6. Clonogenic Assay 

Cells were exposed to SS smoke extract for 48 h, followed by an additional 24 or 48 h with SS smoke extract plus 2.5 µM cisplatin. Cells treated with cisplatin in the absence of SS smoke extract were used as control. After treatment, cells were trypsinized, counted, seeded at very low density, and grown in standard conditions (no treatment) for about 3 weeks. After colony formation, cells were fixed with methanol, followed by staining with 0.5% crystal violet in 25% methanol for 10 min, washed with deionized water and air dried. Colonies with ≥ 50 cells were counted manually as reported earlier [19] and normalized against plating efficiency of control not treated with cisplatin. Experiments were repeated independently at least two times, with each experiment having two independent sets of 3 wells for each cell line and condition. 

### 4.7. Gamma H2AX Assay 

To assess whether exposure to SS extract for up to 96 h increase the DNA double-strand breaks, γH2AX was evaluated by immunofluorescence as per manufacturer’s protocol (Millipore, Burlington, MA, USA). Briefly, cells were plated in 8-well chambered slides, grown in the presence or absence of SS extract for 48 or 96 h and stained for γH2AX. ProLong™ Diamond Antifade Mountant with DAPI (Invitrogen, Waltham, CA, USA) was used to stain nucleus. Data are expressed as the % of cells with γH2AX foci and the number of foci/nuclei in γH2AX-positive cells. Each well was imaged with a minimum of 6 different fields at each time point. Duplicate wells were used for each treatment group. Experiments were repeated independently twice.

### 4.8. Western Blot Analysis 

Cells were exposed to SS smoke extract for 48 h. Unexposed cells were used as control. Protein extraction and Western blot analysis were performed following standard protocols [58]. Briefly, cells were washed with ice-cold PBS, and resuspended in radioimmunoprecipitation assay (RIPA) buffer supplemented with 1X protein inhibitor cocktail and phenylmethylsulfonyl fluoride (PMSF) (Roche, Pleasanton, CA, USA). Then, 40 to 50 µg of protein per sample was loaded onto 10% sodium dodecyl sulfate-polyacrylamide gel electrophoresis (SDS-PAGE). Fractionated proteins were transferred onto polyvinylidene difluoride (PVDF) membranes (Millipore, Rockford, MA, USA). Membranes were blocked with 5% non-fat milk, followed by incubation at 4 °C overnight with primary antibody (Appendix A), standard washing, and incubation with secondary antibodies for 1 h at room temperature. Protein bands were visualized using chemiluminescence reagents (Pierce Biotechnology, IL, USA) and the ChemiDocTM touch imaging system (BioRad, Hercules, CA, USA), and analyzed using Image lab software (BioRad, Hercules, CA, USA). 

### 4.9. RNA Extraction, cDNA Synthesis, and qPCR 

Cells were exposed to SS smoke extract for 48 h. Unexposed cells were used as control. Total RNA was isolated using TRIzol reagent (Invitrogen, Waltham, CA, USA), and converted into complementary DNA (cDNA) using Reverse Transcriptase PCR (Invitrogen, Waltham, CA, USA) following manufacturer’s recommendations. First-strand cDNA was amplified using KAPA SYBR FAST Universal (KAPA Biosystems, Wilmington, MA, USA) and specific primers (Appendix A) on a BioRad CFX384 Real-Time System (C1000 Touch Thermal Cycler, Hercules, CA, USA). β-actin was used as an internal control. 2^−ΔΔCT^ method was used to analyze the relative quantification. Three experiments with 4–6 replicates were performed. 

### 4.10. Statistical Analysis 

The differences in cell death, viability, proliferation, colony survival, gene expression, and protein expression were tested using two-sample independent *t*-tests. All data are presented as mean ± SD, unless otherwise specified. Linear regression analysis was performed to assess the effect of SS smoke exposure on each outcome (cell death, cell viability, and colony survival) after adjusting all three cell lines’ data points as sum. A *p*-value < 0.05 was considered statistically significant. The analysis was performed using SAS 9.4, and GraphPad prism 9.0 software was used to plot IC_50_ curves.

## Figures and Tables

**Figure 1 ijms-25-01032-f001:**
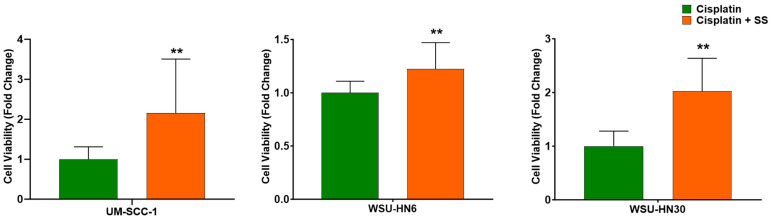
SS smoke extract increases the cell viability of cisplatin-treated head and neck squamous cell carcinoma cells. Cell viability following cisplatin treatment (10 µM) was significantly higher in cells exposed to SS smoke extract than in unexposed cells for all cell lines. Data are shown as mean ± SD. ** *p* < 0.01.

**Figure 2 ijms-25-01032-f002:**
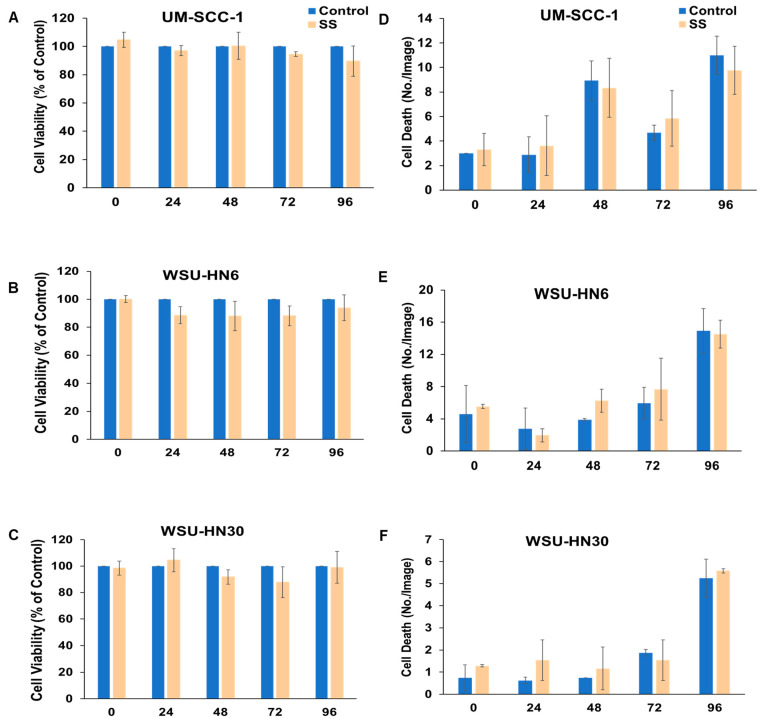
SS smoke extract alone does not alter head and neck squamous cell carcinoma cell viability or cell death. There was no difference in cell viability (**A**–**C**) or in cell death (**D**–**F**) between SS-only-exposed and unexposed cells for any of the cell lines tested. Cell viability was assessed using MTT at 0, 24, 48, 72, and 96 h. Cell death was assessed using Incucyte cytotox red dye at 0, 24, 48, 72, and 96 h. Data are shown as mean ± SD.

**Figure 3 ijms-25-01032-f003:**
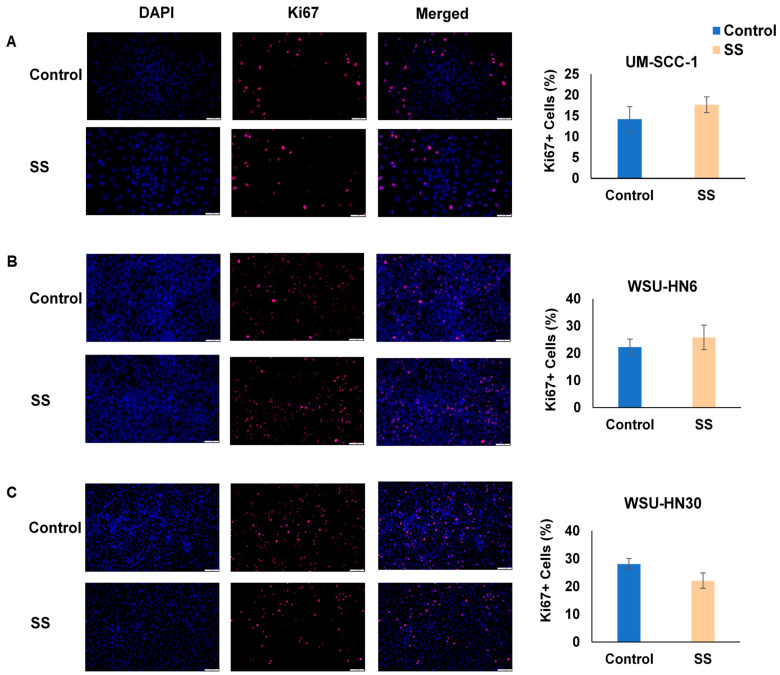
SS smoke extract alone does not alter cell proliferation in head and neck squamous cell carcinoma. There was no difference in cell proliferation between SS-only-exposed and unexposed cells for any of the cell lines tested (**A**–**C**). At 96 h of exposure, cells were stained with Ki67 antibodies (bright, far-red, fluorescent dye) and DNA labelled with DAPI (blue). Data are shown as mean ± SD. Scale bar is 50 µm.

**Figure 4 ijms-25-01032-f004:**
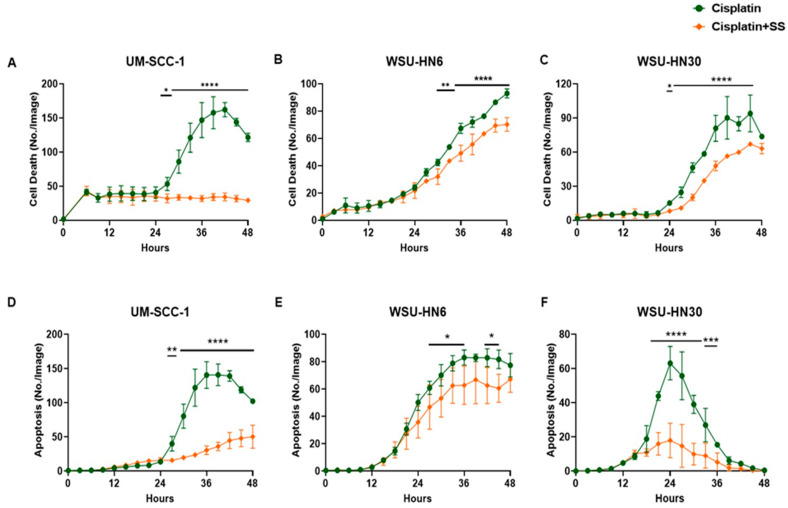
SS smoke extract reduces cisplatin-induced cell death and apoptosis in head and neck cancer cells. HNSCC cells were treated with cisplatin (10 µM) in the presence or absence of SS smoke extracts. Cell death and apoptosis were assessed using Incucyte cytotoxic and Caspase-3/7 dye, respectively. Cancer cells treated with cisplatin in the presence of SS smoke extract showed significantly lower cisplatin-induced cell death (**A**–**C**) and significantly lower apoptosis (**D**–**F**) in all the cell lines tested. Data are shown as mean ± SD. * *p* < 0.05, ** *p* < 0.01, *** *p* < 0.001, **** *p* < 0.0001.

**Figure 5 ijms-25-01032-f005:**
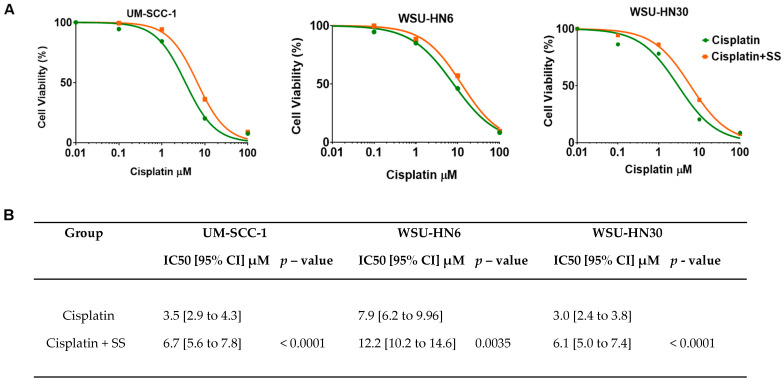
SS smoke extract increases cisplatin resistance in head and neck cancer cells. HNSCC cells were treated with diverse concentrations of cisplatin (0–100 µM) in the presence or absence of SS smoke extracts, and cell viability was measured using MTT. SS smoke extract exposure significantly decreased cisplatin sensitivity and doubled the concentration of cisplatin needed to achieve the IC_50_ values of cisplatin-alone treated cells. (**A**) Dose-dependent symmetrical sigmoidal curves; (**B**) IC_50_ values with 95% CI for each treatment and *p*-values between treatment groups for all the cell lines tested.

**Figure 6 ijms-25-01032-f006:**
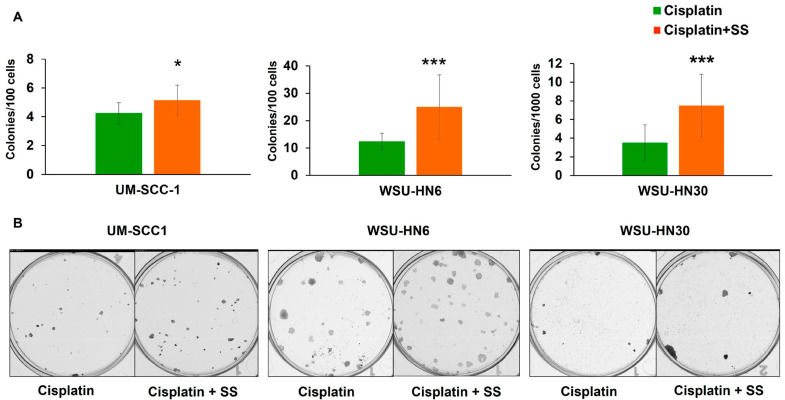
SS smoke extract exposure during cisplatin treatment increases head and neck cancer clonogenic survival. The ability of a cell to proliferate indefinitely was assessed by counting the number of cells forming colonies after 48 h of treatment with cisplatin (2.5 µM) in the presence or absence of SS smoke extract. (**A**,**B**) Colony formation after 48 h of treatment with cisplatin. Data show that the presence of SS smoke extract significantly increases colony formation in all the cell lines. Data are shown as mean ± SD. * *p* < 0.05, *** *p* < 0.001.

**Figure 7 ijms-25-01032-f007:**
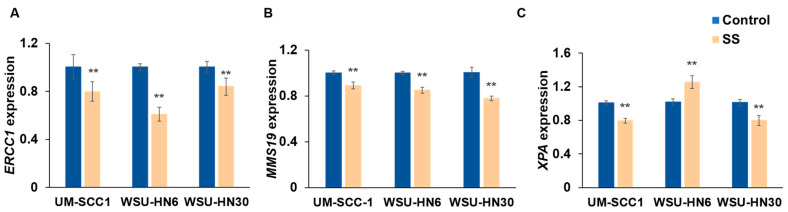
SS smoke exposure alters the expression of nucleotide excision repair genes in head and neck cancer cells. Cells were exposed to SS smoke extract for 48 h and gene expression was assessed by qPCR. Compared to unexposed control cells, exposure to SS smoke extract significantly reduced the expression of (**A**) *ERCC1* and (**B**) *MMS19* genes in all three cell lines. Exposure to SS smoke extract decreased (**C**) *XPA* expression in UM-SCC-1 and WSU-HN30 cells, whereas it increased in WSU-HN6 cells. Data are shown as mean ± SEM. ** *p* < 0.01.

**Figure 8 ijms-25-01032-f008:**
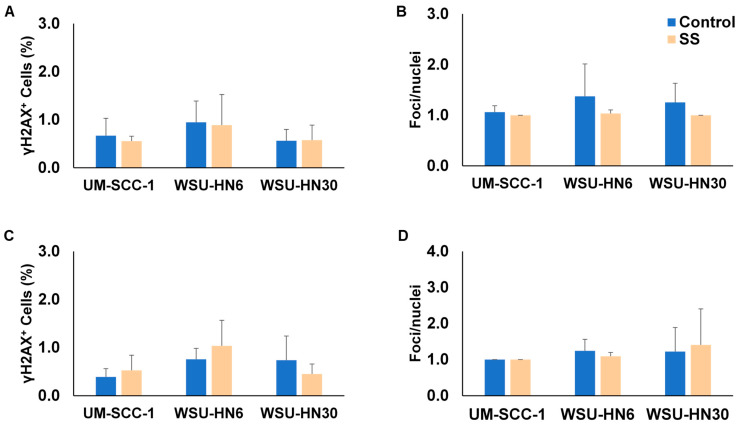
SS smoke extract exposure does not increase double-strand breaks in HNSCC cells. UM-SCC-1, WSU-HN6, and WSU-HN30 cells were exposed to SS smoke extract for 48 h (**A**,**B**) or 96 h (**C**,**D**). Double-strand breaks were quantified using γH2AX immunofluorescence staining assay. No difference was observed between SS-smoke-extract-only-exposed group and unexposed control cells tested at either 48 h or 96 h. Data are shown as mean ± SD.

**Figure 9 ijms-25-01032-f009:**
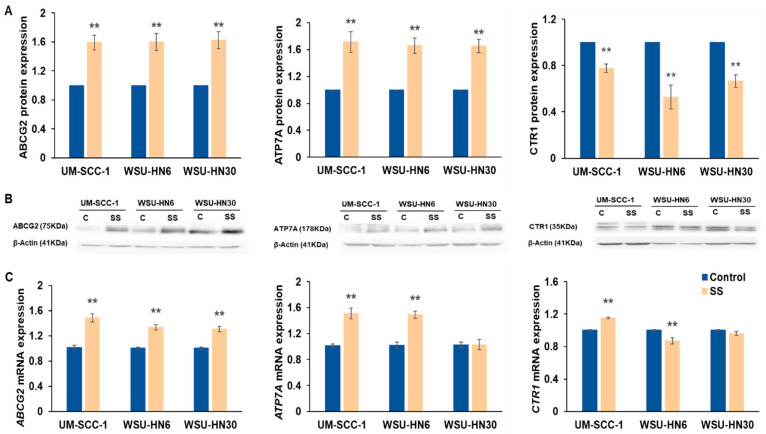
SS smoke exposure alters the expression of cisplatin drug transporters. HNSCC cells were exposed to SS smoke extract for 48 h. (**A**,**B**) ABCG2, ATP7A, and CTR1 protein expression was assessed by western blot analysis. (**C**) *ABCG2*, *ATP7A*, and *CTR1* gene expression was assessed by qPCR. Compared to unexposed control cells, SS smoke extract exposure significantly increased the expression of both ABCG2 and ATP7A at protein level in all the cell lines tested. Additionally, CTR1 protein expression was decreased in all the cell lines tested, whereas *CTR1* mRNA was decreased only in WSU-HN6 cells. Data are shown as mean ± SEM. ** *p* ≤ 0.01.

## Data Availability

All data generated or analyzed during this study are included in this published article.

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
