# Peer review of "Exposure to Secondhand Smoke Extract Increases Cisplatin Resistance in Head and Neck Cancer Cells"

_ijms, 2024, doi:10.3390/ijms25021032_

Round 1
Reviewer 1 Report (New Reviewer)
Comments and Suggestions for Authors
This paper has potential to be published in my opinion and is based in the Chemotherapy resistance are major obstacles in the long-term efficacy of head and neck squamous cell carcinoma (HNSCC) treatment. Furthermore, Secondhand smoke (SHS) exposure is common and has been proposed as an independent predictor of HNSCC recurrence and disease-free survival. Thus, is important to study in vitro using HNSCC cell lines exposed to SHS and clarify mechanisms involved in this response.
I found the paper to be overall well written. However, it needs some adjustments and responses to be published. Therefore, I recommend that a revision is warranted. I explain my concerns in more detail below. I ask that the authors specifically address each of my comments in their response.
The aim of this paper is very short. I believe that the authors could be increase the aims to clarify to readers.
Results section Figure 1. The standard deviation WSU-HN6 WSU-HN30 in condition Cisplatin + SS is much higher. How could the authors explain this?
The phrase in lines (131-132) “Linear regression analysis adjusting for cell line, further strengthen our observation that SS smoke exposure decreased apoptosis across all the cell lines studied (p<0.0001)”. The authors could explain, are where the results?
Lines (154-156) “Since cisplatin IC50 is an in vitro marker predictive of cisplatin response in cancer patients. Since cisplatin IC50 is an in vitro marker predictive of cisplatin response in cancer patients, these results suggest that SHS exposure during HNSCC patient’s cisplatin treatment may reduce treatment efficacy, which in turn could lead to poor prognosis. Rewrite this phrase I believed is not correct sentence to inform this information. Would be possibly change this phrase to discussion session.
The conclusion of this paper is poor in regarding the results showed. The authors should improve this section.
In material and Methods in 4.3. Cell Viability and Cell Proliferation Analysis section, how many times the experiments were done? How many were the replicates?
Author Response
Please see attachment

Reviewer 2 Report (New Reviewer)
Comments and Suggestions for Authors
Dear Authors,
It was a pleasure to read your article. I believe your paper might be interesting to readers from the clinical field. Your paper is well written and organized.
However, there are some scopes to improve the quality of the manuscript. The reviewer would like to suggest the following revision in the manuscript.
The aim of this technical note "Exposure to secondhand smoke extract increases cisplatin resistance in head and neck cancer cells" was to investigate the effects of seckendhand smoke (SHS) exposure on cisplatin efficacy and the potential mechanisms responsible for the observed effects.
English language fine. No issues detected.
Punctuation should be corrected.
Write the article impersonally. Not using the first person plural. Redraft the article.
Standardize text structure and alignment according to guidelines.
Remove academic degrees from the author list.
Add each affiliation separately.
Incorrect citation style in the article. Correct.
Introduction
The introduction is too short. Describe in detail the role of smoking in HNSCC.
Describe why HNSCC is very difficult to treat, such as loss of aesthetics, social roles, speaking, breathing, eating, etc.
Describe the purpose of the work more clearly.
Results
Prepare tables according to the guidelines.
Write p value in italics. Correct in whole article and on figures, e.g. Fig. 5B
The figures are blurry. Enlarge.
Discussion
No described limitations in the discussion. Add the advantages of the study.
Expand as a field for further research.
Materials and methods
Add the names of cities and countries next to the names of products and reagents.
The study should have the approval of the bioethics committee. Not included in this article. Must be completed!
Add a table with abbreviations.
Prepare references according to MDPI guidelines.
Reconsider after major revision
Round 2
Reviewer 2 Report (New Reviewer)
Comments and Suggestions for Authors
The authors followed these guidelines.
Acceptance of the work after the Editor's decision.
This manuscript is a resubmission of an earlier submission. The following is a list of the peer review reports and author responses from that submission.
Round 1
Reviewer 1 Report
Comments and Suggestions for Authors
The manuscript of Dr. Manyanga and collaborators is focused on the investigation of the role of secondhand smoke (SHS) on cisplatin resistance in head and neck cancer. To this extent, the authors treated three HNSCC cell lines (one of which is known to be sensitive to cisplatin) with cisplatin in the presence or absence of an SHS extract and evaluated cell viability, cell death, and some molecular markers associated with platinum resistance. They found that SHS enhanced cisplatin resistance via transcriptional and protein modulation of some genes. Although well written, the manuscript is limited by the lacking of relevant information and by some flaws in the experimental design. Authors are advised to solve the issues listed and detailed below in order to consider their manuscript for publication in Int J Mol Sci journal.
Major issues
1) In order to understand the effects of SHS exposure on cisplatin resistance, authors should include SHS-only treatment (without cisplatin) to determine if the phenomena observed in presence of cisplatin and SHS could be imputed to that combination or just to the SHS action.
2) Authors claim to have measured cell viability and cell proliferation by means of the same assay (MTT assay), which provides information on the cellular metabolic activity. Since they did not provide sufficient information, we can argue that they measured cell viability (as a percentage of untreated cells? See next comment) at 48 hours post-treatment and cell proliferation as absorbance (why did they call it cell viability?) at sequential time points (0, 24, 48, 72 and 96 hours post-treatment). How did the viability of cells change as compared to untreated cells at each time point? It is not clear the reason why the authors did not indicate the findings reported in Supplementary Figure 1 as cell viability (percentage as in Figure 1) for each time point. This leads to some confusion in the data interpretation. In addition to these critical issues, in order to properly refer to cell proliferation, especially in cancer biology, an analysis of specific proliferation markers (such as Ki67, PCNA, etc.) would be advisable.
3) In figure 1A (referred to as ‘cell viability’ by authors) the values are expressed as a percentage of something: no indication is provided, just the reference of a previous work where authors expressed viability as a percentage of untreated cells. It is the same in this manuscript? It must be clearly indicated. Moreover, even if we could hypothesize that C stands for cisplatin-treated cells and SS for cisplatin+SS extract, it must be clearly indicated, at least, in the figure captions.
4) Several biological processes and genes are regulated by the enhanced DNA repair mechanism of platinum resistance (NER, homologous recombination, non-homologous end joining), as pointed out in PMID:34645978. Which was the authors’ rationale to restrict their investigation to NER mechanisms? This point needs to be accurately discussed in the manuscript. Still on the downregulation of NER genes expression, does the SHS-treatment alone, without cisplatin, induce such modulation of these genes? This analysis should be included in the authors' manuscript to expand their investigation and strengthen/support their conclusions.
Minor points
a) The first statement of paragraph 2.3 should be revised. It correctly describes the recurrence phenomenon, but there are some other processes that should be considered and mentioned before considering metastatization process.
b) The re-arrangements of blot lanes do not allow a full evaluation and understanding of experiment quality and should be avoided or supported with full blotting analyses.
Reviewer 2 Report
Comments and Suggestions for Authors
This manuscript addresses an intriguing concept, that exposure to secondhand smoke confers resistance to cisplatin in head and neck cancer. Three head and neck cancer models are utilized for this work, and the experimental approaches are generally appropriate and rigorous.
Unfortunately, the alterations in sensitivity, based on IC50 values, while statistically significant, are quite modest.The change in sensitivity observed is unlikely to be sufficient to substantially influence the clinical response to cisplatin.
Minor point: The data points in Figure 1C should include error bars.
The mechanistic studies are incomplete.. The authors provide data showing reduced expression of nucleotide excision repair genes, but no evidence of increased DNA damage. The authors provide data indicative of increased expression of multidrug resistance proteins, but neglect to access intracellular levels of cisplatin or cisplatin-DNA adduct formation. No studies are provided of drug induced cell killing (via apoptosis or other pathways).
If multiple factors such as drug transport/accumulation and DNA damage/repair were actually altered, one might have expected much more substantial impact on drug sensitivity.